# A Probeless Capacitive Biosensor for Direct Detection of Amyloid Beta 1-42 in Human Serum Based on an Interdigitated Chain-Shaped Electrode

**DOI:** 10.3390/mi11090791

**Published:** 2020-08-21

**Authors:** Hien T. Ngoc Le, Jinsoo Park, Sungbo Cho

**Affiliations:** 1Department of Electronic Engineering, Gachon University, Seongnam-si, Gyeonggi-do 13120, Korea; ltnh1809@gachon.ac.kr; 2Department of Health Sciences and Technology, GAIHST, Gachon University, Incheon 21999, Korea; jspark88@gc.gachon.ac.kr

**Keywords:** interdigitated chain-shaped electrode, anti-aβ antibody, amyloid-β 1-42 peptide, self-assembled monolayer, capacitive biosensor, electrochemical impedance spectroscopy

## Abstract

Amyloid beta (aβ) 1-42, a peptide that is 1-42 amino acids long, is a major component of senile plaques in the brains of patients with Alzheimer’s disease. Aβ detection has become an essential antecedence to predict the declining mental abilities of patients. In this paper, a probeless capacitive biosensor for the non-Faradaic detection of aβ 1-42 peptide was developed by immobilizing a specific anti-aβ antibody onto a self-assembled monolayer functionalized interdigitated chain-shaped electrode (anti-aβ/SAM/ICE). The novelty and difference of this article from previous studies is the direct detection of aβ peptide with no redox probe ((Fe(CN)_6_)^3−/4−^), which can avoid the denaturation of the protein caused by the metallization (binding of aβ to metal ion Fe which is presented in the redox couple). The direct detection of aβ with no redox probe is performed by non-Faradaic capacitive measurement, which is greatly different from the Faradaic measurement of the charge transfer resistance of the redox probe. The detection of various aβ 1-42 peptide concentrations in human serum (HS) was performed by measuring the relative change in electrode interfacial capacitance due to the specific antibody-aβ binding. Capacitance change in the anti-aβ/SAM/ICE biosensor showed a linear detection range between 10 pg mL^−1^ and 10^4^ pg mL^−1^, and a detection limit of 7.5 pg mL^−1^ in HS, which was much lower than the limit of detection for CSF aβ 1-42 (~500 pg mL^−1^) and other biosensors. The small dissociation constant *K_d_* of the antibody-antigen interaction was also found to be 0.016 nM in HS, indicating the high binding affinity of the anti-aβ/SAM/ICE biosensor in the recognizing of aβ 1-42. Thus, the developed sensor can be used for label-free and direct measurement of aβ 1-42 peptide and for point-of-care diagnosis of Alzheimer’s disease without redox probe.

## 1. Introduction

Alzheimer’s disease (AD) is the most common among neurodegenerative brain diseases. Features specific to AD pathology include the atrophy of neurons, synapse loss, and accumulation of senile plaques. These plaques consist of amyloid beta (aβ) peptides and intracellular neurofibrillary tangles (NFTs), containing hyperphosphorylated tau protein [1,2]. Magnetic resonance imaging (MRI) and positron emission tomography (PET) are the diagnostic methods used to predict the stage of AD pathology in clinical practice [3,4,5,6]. However, PET has poor spatial resolution and artifacts of movements, and MRI has low scanning velocity and motion artifacts [7]. Furthermore, these are costly and can have disagreeable activations such as queasiness, megrim, vomiting, fulminate and itching. The detection of biomarkers in cerebrospinal fluid (CSF) is a replacement diagnostic approach. Recently, CSF studies have shown increased levels of tau and phosphorylated tau (p-tau) proteins, and decreased levels of abnormal aβ 1-42 (a main of pathological proteins of AD) [8,9,10]. However, the study of CSF biomarkers is an invasive procedure requiring a lumbar puncture, resulting in back pain [11,12]. Inexpensive and possible methods are thus required for the early detection of aβ to manage AD.

Blood aβ (serum, plasma) has recently been reported as an AD-signaling biomarker, since it can penetrate an endothelial highly specialized membrane that lines cerebral micro-vessels easily into the blood-brain barrier (BBB), that constitutes the interface of the neural cell to the circulating cells of the immune system. According to the amyloid hypothesis in AD, aβ is transported from the brain over the BBB to the blood, through low-density lipoprotein receptor-related protein-1, allowing the clinical use of blood aβ biomarker [12,13,14,15]. The collection and analysis of blood biomarkers are also simple. Although blood-based biomarkers are easy to use and less invasive for the early diagnosis of AD, not much research has been done on these [16,17]. Therefore, new studies on blood serum or plasma biomarkers are essential in the early diagnosis and treatment of AD.

As a valuable tool of sensing biomarkers for the fast, sensible, and selective detective capabilities of aβ at pg mL^−1^ stage [18,19,20,21,22,23,24,25], electrochemical biosensors based on changes in the electrical properties of the electrode surface like capacitance or impedance are proposed. The high sensing performance of biomarkers in a very small sampling volume is strongly related to using nano- or micro-interdigitated electrodes, as a sensing region comparable to the size of the analyte can be adjusted by reducing the distance between the interdigitated electrodes [26,27,28]. Within the typical rectangular-shaped interdigitated electrode, however, the sensing region is non-homogeneous, because the electrical fields are strongly localized on the edge band. Through correctly designing the electrode shape, the edge influence of electric field distribution will be avoided, leading to enhance sensor area homogeneity on electrodes and electrochemical sensor preciseness.

In this paper, we designed a new chain-shaped electrode, to avoid the edge effect of the electric field distribution, and developed a highly sensitive and probeless capacitive biosensor for the non-Faradaic detection of aβ 1-42 peptide in human serum (HS) at different concentrations. In contrast with the Faradaic detection of aβ (with the presence of a redox probe couple (Fe(CN)_6_)^3-/4-^ in phosphate buffer saline), using the change of the charge-transfer resistance as a parameter for the detection in the recent publication [27], the non-Faradaic detection of aβ is the direct detection in only phosphate buffer saline (PBS, pH 7.4) solution (without the addition of any redox probe couple (Fe(CN)_6_)^3−/4−^) is the sensing mechanism, and the change in interfacial capacitance (dielectric layers at the electrode/solution interface) [25] at a single frequency was used as a parameter for detection of aβ in this report, the comparison of 2 sensing mechanism for non-Faradaic and Faradaic detection was shown in Figure 1 and Figure 2. By removing the redox probe couple (Fe(CN)_6_)^3−/4−^ in the measurement, we can avoid the binding of aβ peptide to metal ion Fe which is presented in the redox couple, forming the metal ion complexed aβ which is the cause of the denaturation of aβ peptide, giving disadvantage issues during the measurement [29,30,31]. Moreover, the experimental process become simpler and more suitable for point of care diagnosis without the application of redox probe. The aβ 1-42 could be identified in HS via the biosensor, after a particular anti-aβ antibody was immobilized on a self-assembled monolayer functionalized with an interdigitated chain-shaped electrode (ICE). The developed biosensor provides a linear range of detection from 10 to 10^4^ pg mL^−1^, and a limit of detection of 7.5 pg mL^−1^, which is much lower than the critical concentration value (~500 pg mL^−1^) of CSF aβ 1-42; the value for other assays is shown in Table 1. The selectivity of the biosensor was also elucidated, and no significant interference from C-reactive protein, tumor necrosis factor-alpha, and insulin was observed.

## 2. Materials and Methods

### 2.1. Materials

FDA-approved human serum (HS, H6914 from male AB clotted whole blood, Saint Louis, MO, USA origin, sterile-filtered), bovine serum albumin (BSA), N-hydroxysuccinimide (NHS, 98.0%), N-(3-dimethylaminopropyl)-N′ ethylcarbodiimide hydrochloride (EDC, crystalline), 6-Mercaptohexanoic acid (MHA, 90%) are bought from Sigma-Aldrich (Seoul, Korea).

Phosphate buffered saline (PBS, pH 7.4) is obtained from Tech and Innovation. Anti-aβ antibody MOAB-2, amyloid-β 1-42 (aβ 1-42, human) peptide are obtained from Abcam (Seoul, Korea). De-ionized (DI) water (18.2 MΩ·cm) is taken from the Milli-Q system (Seoul, Korea) and is used for all experiments.

### 2.2. Manufacture Procedure of Interdigitated Chain-Shaped Electrode (ICE)

The ICE was built on a glass diaphragm substrate (13.5 × 16.0 × 0.5 mm). The electron beam evaporator was used to deposit an electrode thickness of 25 nm with titanium and 50 nm with gold. The coupled electrode finger was then shaped by the lifting progress, with a distance and width of 5 μm for working and reference electrodes. Figure 1 illustrates a microscopic ICE picture that contains golden fingers.

### 2.3. Biosensor Construction

In order to extract unclean substances from the electrode surface, bare ICE is washed with ethanol 100%, DI water, and parched under an N_2_ gas flow. MHA (50 mM) is incubated for 12 h and formed a self-assembled monolayer (SAM) on a cleaned gold surface of the ICE. 75 mM of EDC solution and 5 mM of NHS solution for antibody binding subsequently immerse the SAM-modified electrode. Then, ten anti-aβ antibody microliters (100 μg mL^−1^) are dripped down to the modified electrode, and then stored for 1 h in a wet chamber, to prevent the surface from drying during binding. A coupling reaction between the amino group of anti-aβ antibodies and the EDC/NHS-activated MHA molecules on the modified electrodes immobilized the anti-aβ antibody. At the final step, a nonspecific adsorption blocking agent, BSA (0.5% in 1× PBS, pH 7.4), is used to finalize the anti-aβ/SAM/ICE biosensor for aβ 1-42 peptide detection.

To confirm the elemental content of the SAM-modified electrode (SAM/ICE), energy-dispersive X-ray spectroscopy (EDS, S-4700, HITACHI, Tokyo, Japan) was used in this paper.

### 2.4. Electrochemical Impedance Spectroscopy (EIS)

Electrical impedance spectroscopy (EIS) (EC-Lab, Sp-200, Bio-Logic Science Instruments, Seyssinet-Pariset, France) was used for the recording of impedance and capacitance of the biosensor in PBS (pH 7.4). The EIS output signal is measured by applying an input voltage of 10 mV in a frequency range from 100 mHz–100 Hz, to the working and reference electrode.

In the detection in PBS, different aβ concentrations (10^−3^–10^3^ ng mL^−1^) have been incubated in the anti-aβ/SAM/ICE biosensor at room temperature for 20 min, and then the EIS measurement is started. The biosensor washed with DI water and PBS before recording an EIS after each concentration measurement.

To explore the clinical application, the capacitive anti-aβ/SAM/ICE biosensor was employed for the direct detection of aβ 1-42 peptide in HS. Firstly, the pure HS was diluted in 1× PBS buffer at the ratio (1:1000), to avoid matrix effects in the measurement. Next, 1 μg mL^−1^ of aβ 1-42 in pure HS was made, and then differential concentrations of aβ from 10^−3^ to 10^3^ ng mL^−1^ were diluted using HS (1:1000). Various concentrations of aβ (10^−3^–10^3^ ng mL^−1^) were incubated in the HS (1:1000) with the biosensor for 20 min, and the capacitance was measured.

## 3. Results and Discussion

### 3.1. Electrochemical Characterization of the Anti-aβ Antibody onto a Self-Assembled Monolayer Functionalized Interdigitated Chain-Shaped Electrode (Anti-aβ/SAM/ICE) Biosensor

Construction of the biosensor along with the immobilization of SAM and anti-aβ on the gold surface of ICE, as described in detail in Section 2.3, is shown in Figure 1. The comparison between the non-Faradaic detection and Faradaic detection of aβ [27] was shown in Figure 2, indicating two differential sensing mechanism in the absence and presence of redox probe ((Fe(CN)_6_)^3−/4−^. The direct detection of aβ with no redox probe is performed by non-Faradaic capacitive measurement, which is greatly different from the Faradaic measurement of the charge transfer resistance of the redox probe. The novelty and difference of non-Faradaic capacitive measurement in this report, as compared to the impedance detection of the charge transfer redox probe couple, in previous studies including our article [18,20,27,33], is the direct detection of aβ with no redox probe, which can avoid the denaturation of protein caused by the metallization (binding of aβ to metal ion Fe which is presented in the redox couple) [29,30,31].

EDS was used to confirm the elemental content of the electrode surface, along with the SAM deposition. The most studied of SAM indicated that the binding between the sulfur atom of SAM and gold is very strong and stable [34]. Figure 3a,b show the EDS results of bare ICE and SAM/ICE, respectively; the sulfur (S) element appeared at a small peak of 2.3 keV in Figure 3b, as compared to the absence of S in the Figure 3a, corresponding to the EDS of S [35] and the atomic percentage (at%) of S was found to be 5 at% after SAM deposition as shown in Figure 3b, demonstrating that the gold surface of ICE was successfully modified by SAM via the strong affinity of sulfur for gold.

To confirm SAM deposition, and the binding of anti-aβ, and aβ to the electrode surface, impedance and capacitance was measured in PBS (pH 7.4), over a frequency range of 100 Hz–100 mHz. As shown in Figure 4a, the impedance of the electrode was linearly increased after SAM, anti-aβ, and aβ deposition, respectively; conversely, a decrease in reactive capacitance at low frequency was observed, as shown in Figure 4b, which is characteristic of capacitance from the following of impedance: *C* = 1/(2π*f* × *Z*), where *C* represents capacitance, *f* is the frequency expressed in Hz, and *Z* represents impedance [36]. The decreased capacitance of the biosensor after SAM, anti-aβ, and aβ immobilization was due to the formation of a series of dielectric layers at the electrode/solution interface [37], according to the capacitive series in Figure 1. The capacitance of the sensor at the electrode/solution interface could be depicted to be built-up of several capacitors in series. The first capacitance constitutes the insulating layer as SAM on the electrode surface, *C*_SAM_. The second capacitor, *C*_anti-aβ,_ includes the anti-aβ molecular layer. The third capacitor is defined by the concentration-dependent of aβ 1-42 peptide, *C*_aβ_. The total capacitance of the biosensor after immobilizing SAM, anti-aβ, and aβ 1-42 peptide can be expressed as [38]:1/*C*_total_ = 1/*C*_SAM_ + 1/*C*_anti-aβ_ + 1/*C*_aβ_(1)

The decrease in capacitance (or increase in impedance) with the deposition of SAM indicates that an insulating well-organized structure of long-chain alkanethiols from SAM was formed on the gold surface of ICE, a property important in the construction of capacitive biosensors [34,39].

### 3.2. Capacitive Detection of aβ in PBS by the Biosensor

The capacitive detection of aβ 1-42 in a wide range of concentrations from 10^−3^ to 10^3^ ng mL^−1^ in PBS (pH 7.4), by the anti-aβ/SAM/ICE biosensor, is shown in Figure 5. Changes in the capacitance values of the biosensor at various concentration of aβ were estimated by the normalization of capacitance as a function as Δ*C* = |(*C*_aβ in PBS_ − *C*_0_)/*C*_0_|, where *C*_0_ represents the capacitance of the biosensor with anti-aβ antibody deposition, and *C*_aβ in PBS_ represents the capacitance of the biosensor with aβ (10^−3^–10^3^ ng mL^−1^) incubation in PBS. From the determined values of Δ*C* versus the frequency *f* in Figure 5a, the change or increase in Δ*C* was observed in the increased aβ concentrations, indicating that Δ*C* could be used as a parameter for the sensitive detection of aβ. Therefore, the plot of Δ*C* at a frequency of 1 Hz vs. concentrations of aβ was established, to determine the calibration curve consisting of the linear range (from 10^−2^ to 10^1^ ng mL^−1^) and the saturation range (from 10^2^ to 10^3^), for the detection of aβ in Figure 5b. In the linear range from 10^−2^ to 10^1^ ng mL^−1^ (Figure 5b inset), a calibration curve was established to determine the limit of detection (LOD) of the biosensor in PBS; the LOD was found to be 6.75 × 10^−3^ ng mL^−1^ (6.75 pg mL^−1^) that was calculated by (3S/b), where S is the standard deviation of the intercept and b is the slope of the linear range [40]; and the linear range of detection (LRD) was from 10^−2^ to 10^1^ ng mL^−1^.

### 3.3. Capacitive Detection of aβ in Human Serum (HS) by the Biosensor

The capacitive anti-aβ/SAM/ICE biosensor was used to detect aβ 1-42 peptide in HS, to examine the clinical applications of the biosensor. The experimental procedure for the detection of aβ in HS was described in Section 2.4. The response of the biosensor to aβ at various concentrations from 10^−3^ to 10^3^ ng mL^−1^ in HS is shown in Figure 6. The corresponding values of Δ*C* in Figure 6a were increased with increasing concentrations of aβ, indicating that the biosensor could detect aβ in HS as well as in PBS (as shown in Figure 5). Figure 6b also showed the linear range from 10^−2^ to 10^1^ ng mL^−1^ and the saturation range from 10^2^ to 10^3^ ng mL^−1^. From the linear range from 10^−2^ to 10^1^ ng mL^−1^ of Δ*C* values with concentrations of aβ in HS at 1 Hz of frequency (Figure 6b inset), the LOD was defined as 7.5 × 10^−3^ ng mL^−1^ (7.5 pg mL^−1^), and LRD ranged from 10^−2^ to 10^1^ ng mL^−1^, respectively. The LOD in HS of the capacitive anti-aβ/SAM/ICE biosensor showed a lower value than other aβ sensors that have been proposed recently (Table 1), demonstrating that the biosensor could be used in clinical applications.

### 3.4. Selectivity and Stability

To verify the selectivity of the anti-aβ/SAM/ICE biosensor, the effects of tumor necrosis factor-alpha (TNF-α, 0.5 ng mL^−1^), insulin (10^3^ ng mL^−1^), and C-reactive protein (CRP, 10^5^ ng mL^−1^) in HS were characterized using the change in capacitance Δ*C* value at 1 Hz frequency (Figure 7a). The Δ*C* response of CRP, TNF-α, and insulin did not change much compared to the Δ*C* response for aβ, indicating that there was no significant interference from CRP, TNF-α, and insulin. The change in capacitance was due to the capability of specific antibody-antigen binding, demonstrating that the biosensor was selective for aβ detection. Moreover, the fabricated aβ/SAM/ICE biosensors were kept in the refrigerator at 4 °C for storage. Then, the capacitance of the biosensor at 1 Hz was recorded in PBS at 24 °C of room temperature, on the 1st, 2nd, 3rd, 7th, and 14th day after taking it out to check the stability of the biosensor, as shown in Figure 7b. The capacitance at 1 Hz values measured during 14 days of storage showed low deviation to be 3.89% of RSD, illustrating the stability of the anti-aβ/SAM/ICE biosensor.

### 3.5. Binding Affinity and Dissociation Constant K_d_ of Anti-aβ Antibody—aβ 1-42 Peptide Interaction

Binding affinity is the strength of the binding interaction between an antibody to its antigen. Binding affinity is usually measured and stated by the dissociation constant (*K_d_*), which is used to assess and categorize order strengths of antibody-antigen interactions. The smaller the *K_d_* value, the greater the binding affinity of the antibody for its antigen. The larger the *K_d_* value, the weaker the binding affinity between the antigen and antibody.

In this paper, the dissociation constant *K_d_* for the interactions between antigen and its antibody is given below using the Langmuir-adsorption-model-based approach [41]. The equilibrium of the antigen aβ (A) and antibody (B) bindings can be symbolized as:(AB) → (A) + (B)(2)
*K_d_* = (A)(B)/(AB)(3)

Assuming the surface coverage of the antibody-antigen complex (AB) is *θ*, the surface coverage of the unbound antibody (B) will be 1 − θ, so the *K_d_* is:*K_d_* = ((1 − θ)/θ)(A)(4)

From the Langmuir adsorption model, Δ*C* is assumed, that is directly related to the antibody-antigen binding as:Δ*C* = *θ* Δ*C*_max_(5)
where Δ*C*_max_ is the maximum response of the sensor and equal to |(*C*_aβ max_ − *C*_0_)/*C*_0_|.

From Equations (4) and (5), a linearized form of the Langmuir isotherm equation can be expressed as
(A)/Δ*C* = (A)/Δ*C*_max_ + *K_d_*/Δ*C*_max_(6)
where (A) represents the concentrations of the aβ (Conc_aβ_ (ng mL^−1^)).

Using the Equation (6), two linear regression curves of Conc_aβ_ versus Conc_aβ_/Δ*C* were established in PBS and HS as shown in Figure 8. The slope and y-intercept of regression curves in PBS and HS were found to be 0.89 and 1.11; 0.88 and 1.23, respectively. The dissociation constant *K_d_* of the antibody-antigen binding in PBS and HS was obtained by dividing the y-intercept by the slope, and found to be 1.24 and 1.40 ng mL^−1^, which corresponds to 0.014 and 0.016 nM, respectively. This dissociation constant *K_d_* value is smaller as compared to the obtained *K_d_* of others aβ 1-42–binding partner interactions that have been shown recently in Table 2, indicating the high binding affinity between the antibody and antigen in this report, and also confirming the advance of the anti-aβ/SAM/ICE biosensor in the recognizing of aβ 1-42 peptide.

## 4. Conclusions

In this paper, we improved a probeless, label-free, directly and highly sensitive capacitive biosensor for the non-Faradaic detection of the aβ 1-42 peptide, one of main biomarkers of AD in HS. The biosensor was developed simply by immobilizing a specific anti-aβ antibody onto a SAM functionalized interdigitated chain-shaped electrode designed to improve the sensing area homogeneity. By measuring the change in capacitance at the electrode/solution interface without the redox probe couple, the developed biosensor could avoid the denaturation of the protein caused by the metallization between aβ and ion Fe in redox probe, and directly detect aβ in a wide range of 10–10^4^ pg mL^−1^ with a low LOD of 7.5 pg mL^−1^ in HS, which was much lower than the LOD of CSF aβ 1-42 (~500 pg mL^−1^) and other aβ 1-42 biosensors that have been suggested recently. Moreover, our biosensor has several advantages, including high binding affinity of antigen-antibody interaction, small size, light-independent point-of-care diagnosis, preventing the denaturation of target proteins, and good selectivity for aβ 1-42 in HS containing CRP, TNF-α, and insulin, which are the benefits of evaluation criteria for the practical application and the early diagnosis of AD.

## Figures and Tables

**Figure 1 micromachines-11-00791-f001:**
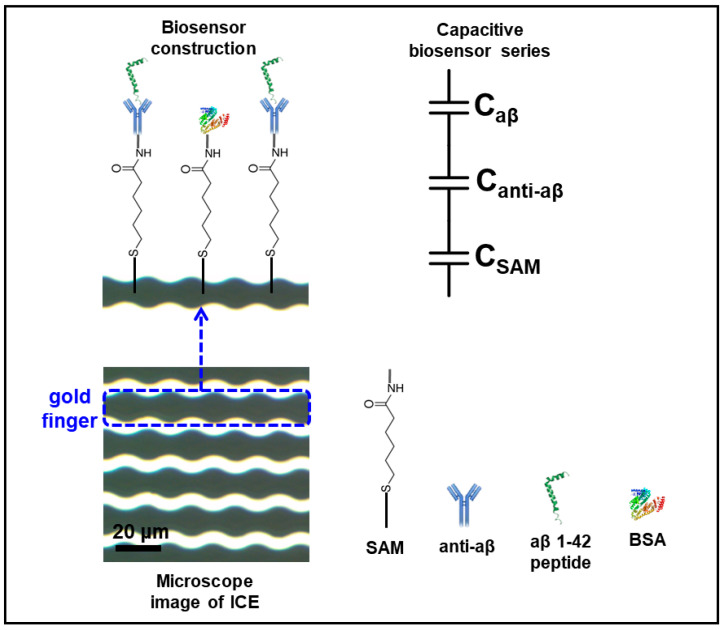
Microscopic image of interdigitated chain-shaped electrode (ICE) and schematic of capacitive biosensor construction.

**Figure 2 micromachines-11-00791-f002:**
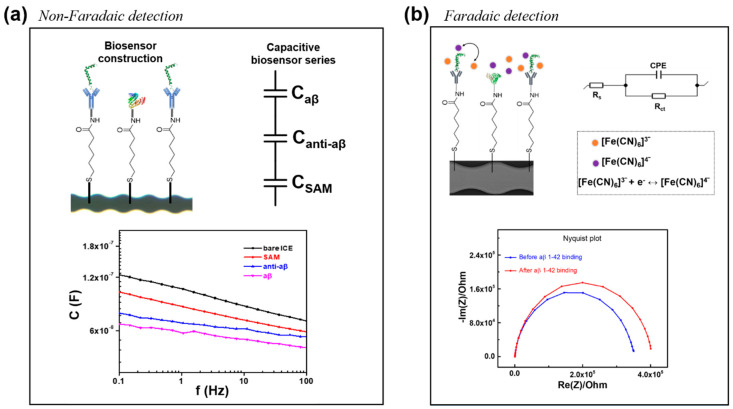
Sensing mechanism for (**a**) Non-Faradaic detection and (**b**) Faradaic detection.

**Figure 3 micromachines-11-00791-f003:**
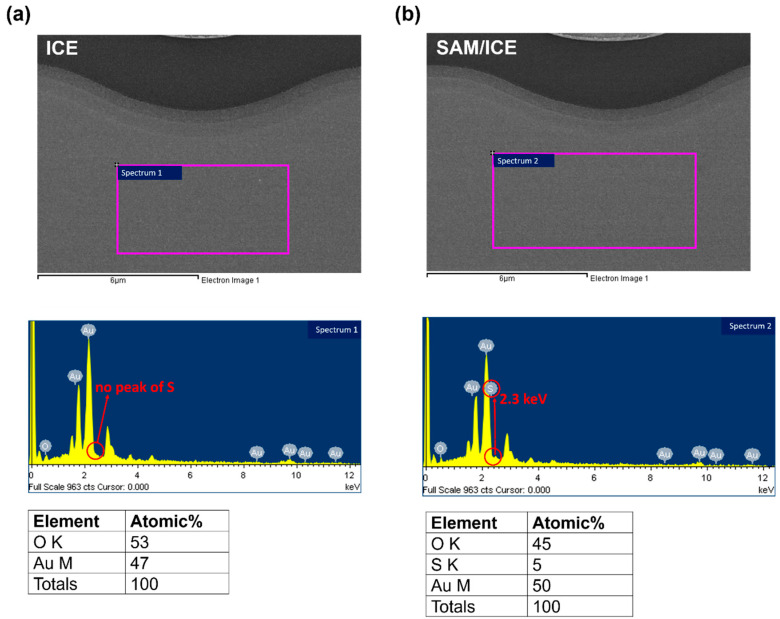
EDS results of (**a**) bare ICE, and (**b**) self-assembled monolayer-modified electrode (SAM/ICE).

**Figure 4 micromachines-11-00791-f004:**
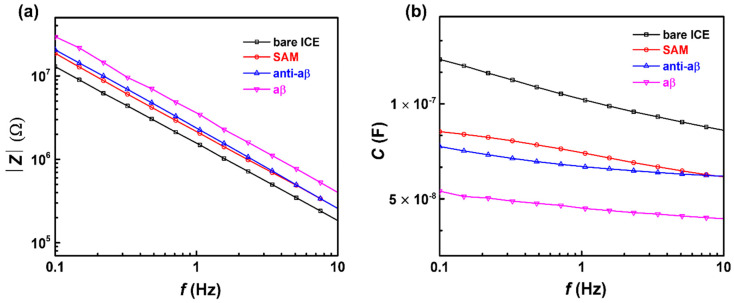
Impedance (**a**) and capacitance (**b**) of biosensor construction at each stage of biosensor modification: bare ICE, SAM, anti-aβ, and aβ in PBS (pH 7.4).

**Figure 5 micromachines-11-00791-f005:**
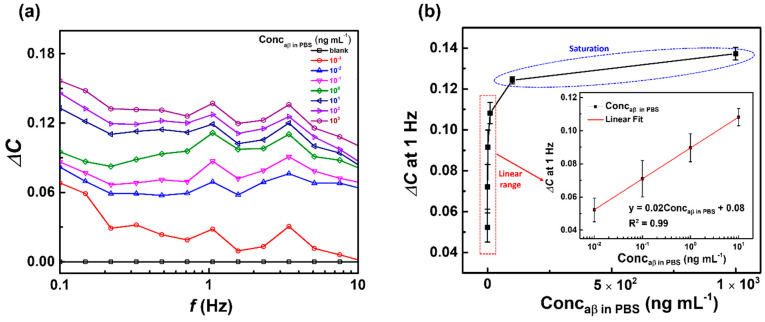
(**a**) Change in capacitance Δ*C*, and (**b**) plot of Δ*C* at 1 Hz vs. concentrations of aβ of the biosensor after incubation with different concentrations of aβ (10^−3^–10^3^ ng mL^−1^) in PBS—inset is the calibration curve of the biosensor in the linear range of aβ concentrations from 10^−2^–10^1^ ng mL^−1^; symbols and bars represent the average and standard deviation of the data (*n* = 3).

**Figure 6 micromachines-11-00791-f006:**
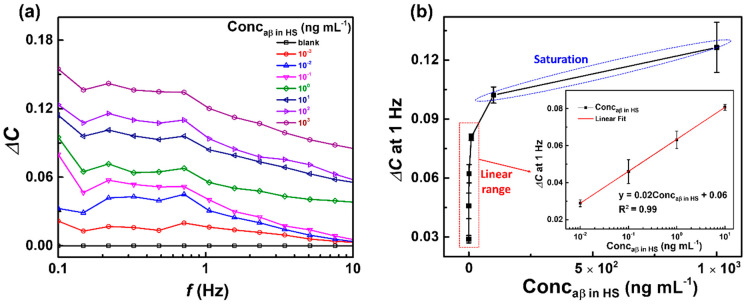
(**a**) Change in capacitance Δ*C*, and (**b**) plot of Δ*C* at 1 Hz vs. concentrations of aβ of the biosensor after incubation with different concentrations of aβ (10^−3^–10^3^ ng mL^−1^) in human serum (HS); inset is the calibration curve of the biosensor in the linear range of aβ concentrations from 10^−2^–10^1^ ng mL^−1^; symbols and bars represent the average and standard deviation of the data (*n* = 3).

**Figure 7 micromachines-11-00791-f007:**
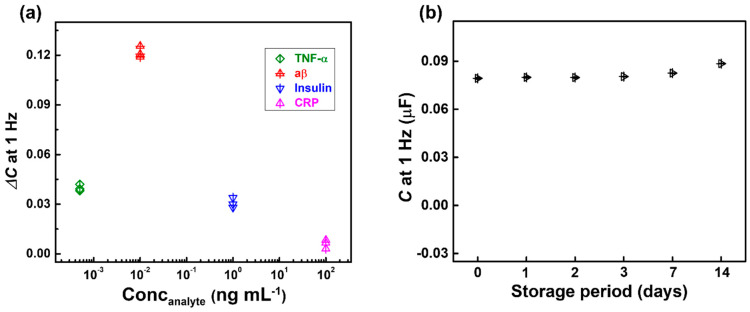
(**a**) Change in capacitance Δ*C* observed at 1 Hz with different analytes to estimate the selectivity of biosensor (*n* = 3), and (**b**) Stability of the biosensor.

**Figure 8 micromachines-11-00791-f008:**
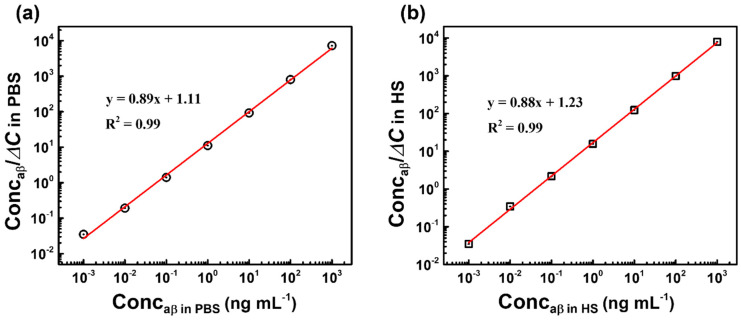
Two plots of C_oncaβ_ versus C_oncaβ_/Δ*C* in (**a**) PBS, (**b**) HS, and fitted with Equation (6) for determining *K_d_*.

**Table 1 micromachines-11-00791-t001:** Electrochemical biosensor for aβ 1-42 detection.

Transducer	Immobilization	Detection	Redox Probe/Electrolyte	LOD (pg mL^−1^)	LRD (pg mL^−1^)	Refs
Carbon disposable electrochemical printed chip	SAM ^a^-AuNPs ^b^	Electrochemical impedance spectroscopy	K_3_[Fe(CN)]_6_/K_4_[Fe(CN)]_6_ in KCl	2.6 × 10^3 ##^	45–4.5 × 10^5^	[18]
Gold electrode	Peptide probe (11-mercaptou-ndecanoic acid + Peptide chain + Ferrocene)/9-mercapto-1-nonanol	Square-Wave Voltammetry	0.1 M KCl containing 5 mM Fe(CN)_6_^3−/4−^	1.1 × 10^3 #^	2.2 × 10^3^–5.4 × 10^4^	[20]
Carbon electrode	AuNPs ^b^/S-AM ^a^ formation of the acetylenyl group/cycloaddition reaction of an azide-terminated sialic acid	Differential Pulse Voltammetry	K_3_[Fe(CN)_6_] in PBS (pH 7.4)	4.5 × 10^6 #^	2.3 × 10^6^–4.5 × 10^7^	[21]
Interdigitate-d chain-shaped electrode	SAM ^a^	Faradaic detection	Fe(CN)_6_^3−/4−^ in PBS (pH 7.4)	100	1–10^6^	[27]
Glassy carbon electrode	Direct immobilization on the electrode surface	Square-Wave Voltammetry	20 mM Tris/HCl buffer, pH 7.0 (TBS)	7.0 × 10^5^	2.8 × 10^6^–1.6 × 10^7^	[32]
Au ^c^ electrode	Microporous Au ^c^ nanostructure/SAM ^a^	Differential Pulse Voltammetry	Fe(CN)_6_^3−/4−^ in KCl	0.2	3–7000	[33]
Interdigitate-d chain-shaped electrode	SAM ^a^	Non-Faradaic detection	No probe	7.5	10–10^4^	This work

^a^ SAM: self-assembled monolayer; ^b^ AuNPs: gold nanoparticles; ^c^ Au: gold; ^#^ Value is expressed in μmol L^−1^ and converted to pg mL^−1^; ^##^ Value is expressed in pM and converted to pg mL^−1^.

**Table 2 micromachines-11-00791-t002:** Dissociation constants *K_d_* of the interaction between aβ 1-42 peptide and various binding partners.

Method	inding Partner	*K_d_* (nM)	Refs
AUC ^a^	antibody 6E10	30.1; 13.2; 63.3	[42]
MST ^b^	antibody 4G8	11.3; 1.0; 45.5	
SPR ^c^	antibody 12F4	14.6; 4.7; 37.1	
Fluorescent	NIAD-4 ^d^	10	[43]
	CRANAD-2 ^e^	38.7	
	DANIR-2c ^f^	26.9	
SPR ^c^	Affibody molecules	1–3	[44]
Capacitive biosensor	antibody MOAB-2	0.014; 0.016	This work

^a^ AUC: analytical ultracentrifugation, ^b^ MST: microscale thermophoresis, ^c^ SPR: surface plasmon resonance, ^d^ NIAD-4: {(50-(p-Hydroxyphenyl)-2,20-bithienyl-5-yl)-methylidene}-propanedinitrile, ^e^ CRANAD-2:(T-4)-((1E,6E)-1,7-bis(4-(dimethylamino)phenyl)-1,6-heptadiene-3,5-dionato-kO^3^,kO^5^)difluoro-boron, ^f^ DANIR-2c: the N,N-dimethylaminophenyl-donating group and dicyanomethylene-accepting group bridged by π-conjugated double bond(s).

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
