# Peer review of "A Probeless Capacitive Biosensor for Direct Detection of Amyloid Beta 1-42 in Human Serum Based on an Interdigitated Chain-Shaped Electrode"

_micromachines, 2020, doi:10.3390/mi11090791_

Round 1

Reviewer 1 Report

In the manuscript by Hien T. Ngoc Le the authors present the probeless detection of amyloid beta 1-42 in human plasma with the use of chain IDEs, through the measurement of capacitance. The authors have recently published an identical paper on their findings https://doi.org/10.1016/j.bios.2019.111694

where impedance was being measured instead of capacitance. The latter was a much more complete paper where the electrodes were satisfactorily characterized and the results more thoroughly described and analyzed. In this manuscript the only new thing the authors present is the conversion of impedance to capacitance. Hence the manuscript does not present any novel findings and I do not recomment its publication. 

Reviewer 2 Report

I

In this paper a label-free capacitive biosensor for non-Faradaic detection of the aβ 1-42 peptide is described. Prior to its publication on this journal I have some comments and suggestions.

In the abstract it is written “…Capacitance change in the antiβ/SAM/ICE biosensor showed a wide linear detection range between 1 pg mL-1 and 1 µg mL-1 , and a detection limit of 150 pg mL-1 …” This statement does not make much sense to me, since the LOD value cannot be higher than the  first point of the linear range. It is necessary review these data.

While checking figure 4, it seems that the calibration curve starts saturating at a concentration level of 10 ng/mL. In my opinion, it might be necessary to measure some intermediate concentrations points between 10 ng/mL and 1000 ng/mL to evaluate the linear range.

What is the difference between the present manuscript and the article Hien T. Ngoc Le et al., Biosensors and Bioelectronics 144 (2019) 111694 already published by the authors? In this very similar paper, the LOD is even better than the achieved in the present manuscript, and it is not even included/mentioned in the present manuscript. 

Taking into account that the cut-off value for aβ 1-42 peptide  is 500 pg/mL and the obtained linear range, why a 1:1000 HS dilution was chosen? In my opinion, if the sample is diluted 1000 times, the concentration of aβ 1-42 peptide would be below the limit of detection of the developed biosensor.  

Table S1 in the supplementary material must be improved. First, reference 2 is repeated and, in my opinion, the literature searching should improved  since articles such as Negahdary et al., Microchim Acta 186, 766 (2019) or Hien T. Ngoc Le et al., Biosensors and Bioelectronics 144 (2019) 111694 were not included. Please, double check literature searching and complete and edit the table accordingly.

Reviewer 3 Report

The authors present and proof an interesting approach to design a selective sensor for the detection of Amyloid Beta peptide. They also demonstrated the selectivity of the biosensor in the case of  C-reactive protein, tumor necrosis factor-alpha, and insulin. The reviewer recommends publishing the paper with minor revisions.

Some discussion is needed for why aβ is available in the blood despite the blood-brain barrier.

The human serum used in the experiments were purchased from Sigma-Aldrich (Korea). Please prove whether the authors need to publish additional info to meet the ethical requirements working with human samples.  

The figures have no uniform description of the units, sometimes /, sometimes in brackets for instance in Figures 5b and 5b; sometimes the units are directly on the axis (Figure 6).

It is unclear what Figure 2 shows. The figure should demonstrate the availability of the sulfur atom on the electrodes. However, this cannot be distinguished directly from the figures presented. What is the reason to show these pictures? In addition, all pictures include the same label “spectrum 1”

The statement is unclear:

“Moreover, the value of capacitance at 1 Hz of the biosensor  (Figure 6 (b)) did not change much after 1, 2, 3, 7, and 14 days of storage at 4oC, illustrating the stability of the anti-aβ/SAM/ICE biosensor.”

What exactly and how was stored?

Round 2

Reviewer 1 Report

The authors have adequately described the novelty in their research methodology compared to their previous published work. Therefore I recommend the publication of this manuscript.